# MicroRNA-223 Suppresses Human Hepatic Stellate Cell Activation Partly via Regulating the Actin Cytoskeleton and Alleviates Fibrosis in Organoid Models of Liver Injury

**DOI:** 10.3390/ijms23169380

**Published:** 2022-08-19

**Authors:** Chaiyaboot Ariyachet, Nattaya Chuaypen, Pornchai Kaewsapsak, Naphat Chantaravisoot, Depicha Jindatip, Saranyapin Potikanond, Pisit Tangkijvanich

**Affiliations:** 1Department of Biochemistry, Faculty of Medicine, Chulalongkorn University, Bangkok 10330, Thailand; 2Center of Excellence in Hepatitis and Liver Cancer, Faculty of Medicine, Chulalongkorn University, Bangkok 10330, Thailand; 3Department of Anatomy, Faculty of Medicine, Chulalongkorn University, Bangkok 10330, Thailand; 4Department of Pharmacology, Faculty of Medicine, Chiang Mai University, Chiang Mai 50200, Thailand; 5Research Center of Pharmaceutical Nanotechnology, Chiang Mai University, Chiang Mai 50200, Thailand

**Keywords:** hepatic stellate cell, liver fibrosis, microRNA, miR-223, cytoskeleton, mechanotransduction, organoid model

## Abstract

MicroRNAs (miRNAs) are small, non-coding RNAs that negatively regulate target mRNA expression, and altered expression of miRNAs is associated with liver pathological conditions. Recent studies in animal models have shown neutrophil/myeloid-specific microRNA-223 (miR-223) as a key regulator in the development of various liver diseases including fibrosis, where hepatic stellate cells (HSCs) are the key player in pathogenesis. However, the precise roles of miR-223 in human HSCs and its therapeutic potential to control fibrosis remain largely unexplored. Using primary human HSCs, we demonstrated that miR-223 suppressed the fibrogenic program and cellular proliferation while promoting features of quiescent HSCs including lipid re-accumulation and retinol storage. Furthermore, induction of miR-223 in HSCs decreased cellular motility and contraction. Mechanistically, miR-223 negatively regulated expression of smooth muscle α-actin (α-SMA) and thus reduced cytoskeletal activity, which is known to promote amplification of fibrogenic signals. Restoration of α-SMA in miR-223-overexpressing HSCs alleviated the antifibrotic effects of miR-223. Finally, to explore the therapeutic potential of miR-233 in liver fibrosis, we generated co-cultured organoids of HSCs with Huh7 hepatoma cells and challenged them with acetaminophen (APAP) or palmitic acid (PA) to induce hepatotoxicity. We showed that ectopic expression of miR-223 in HSCs attenuated fibrogenesis in the two human organoid models of liver injury, suggesting its potential application in antifibrotic therapy.

## 1. Introduction

Liver fibrosis is caused by chronic liver injury and sets the stage for other pathological conditions including cirrhosis, portal hypertension, liver failure, and hepatocellular carcinoma [1]. Given that current antifibrotic therapy is ineffective, understanding the mechanisms of liver fibrosis is crucial to identify new therapeutic targets [2,3]. Upon liver damage, hepatic stellate cells (HSCs) are mainly responsible for the progression of liver fibrosis by transforming their characteristics from a quiescent to an activated state, leading to the excessive production of extracellular matrix (ECM) and formation of a fibrous scar [4,5]. Experimental and clinical data suggest that liver fibrosis could be regressed when the etiological source of chronic injury is removed [6,7]. Thus, molecular control underlying HSC activation and its reversal has been intensively studied to gain new insights for antifibrotic interventions.

Recent studies emphasize the involvement of microRNAs (miRNAs) in HSC activation and their potential as therapeutic targets for liver fibrosis [8,9,10,11]. These small, non-coding miRNAs are a class of endogenous RNAs with ~18–24 nucleotides in length and suppress gene expression by targeting specific messenger RNAs (mRNAs) to induce degradation or hinder the process of translation [12,13]. Comparative expression profiling of miRNAs in human activated and quiescent HSCs reveals miRNA-223 (miR-223) as the most downregulated miRNA upon HSC activation [14]. Intriguingly, genetic ablation of miR-223 exacerbates liver fibrosis in mice upon injury [15,16]. These results suggest miR-223 plays a protective role against fibrogenesis and could represent a new therapeutic target to combat with liver fibrosis [17,18].

X-chromosome-linked miR-223 is highly conserved in vertebrates and specifically expressed in neutrophils and, to a lesser extent, in macrophages, where miR-223 controls neutrophil maturation and macrophage polarization [19,20,21]. A recent report demonstrates that miR-223 can be transmitted via extracellular vesicles (EVs) from neutrophils to hepatic macrophages and promotes the phenotypic conversion of proinflammatory macrophages into a restorative stage that mitigates fibrogenesis by impairing HSC activation [15]. Consistently, lipofectamine-based delivery of synthetic miR-223 into mice with chronic liver injury reduces liver inflammation and improves fibrotic conditions [22]. Other studies suggest the crosstalk between immune cells and HSCs via miR-223-containing EVs which inhibit HSC activation [16,23]. However, due to the complex and dynamic interactions between liver cell types including hepatocytes, Kupffer cells and liver sinusoidal endothelial cells (LSEC) in progression of liver fibrosis [24,25], the anti-fibrotic mechanisms of miR-223 in HSCs remains incompletely understood. In addition, these previous studies rely on analyses from animal models or HSC immortalized cell lines that may not fully reflect biological and clinical relevance for humans. Thus, functional validation of miR-223 in more physiological models of liver fibrosis is warranted to unravel the mechanisms of HSC activation and to explore the therapeutic potential of miR-223 for antifibrotic therapy. The recent development of hepatic organoid models or “mini livers” is an appealing approach to study human liver diseases including fibrosis [26,27,28,29]. These organoids contain one or more liver cell types that mimic in vivo organ architecture and capture complex pathologies ranging from simple steatosis to steatohepatitis, fibrosis, and finally even hepatocellular carcinoma [30,31,32]. This methodology could represent a new avenue for elucidation of pathological mechanisms and identification of new effective treatments.

In this study, we investigated the roles of miR-223 in HSC activation in primary human HSCs, the gold standard to study human liver fibrosis in vitro, and its therapeutic potential in controlling fibrosis using injury models of co-cultured human hepatic organoids.

## 2. Results

### 2.1. Ectopic miR-223 Expression Suppresses Activation of Human HSCs

We established culture of primary human HSCs which appear fibroblastic morphology and show enrichment of HSC-specific markers including retinol binding protein 1 (RBP1) and Gata4 in comparison of dermal fibroblasts [33,34] (Appendix A). Expression of miR-223 is not detectable in cultured HSCs, consistent with the known restriction of miR-223 expression in the myeloid lineage [19,21] (Figure 1a). However, given the fact that miR-223 is found to be a top downregulated miRNA upon culture activation of human HSCs from healthy livers [14], we hypothesized that HSCs could uptake miR-223 from exogenous sources as previously reported [16,23], and restoration of miR-223 levels in activated HSCs may provide antifibrotic effects. To test this hypothesis, we ectopically expressed miR-223 in primary human HSCs via lentiviral gene transfer. This method allows stable expression of the transgene and overcomes a particular problem with primary cells which are often difficult to transfect [35]. Lentivirus carrying the pre-miRNA sequence of miR-223 was transduced into primary HSCs followed by puromycin selection to enrich cells with miR-223 overexpression (dubbed as “miR-223 HSCs”). Primary HSCs transduced with scramble miRNA lentivirus served as a control. We confirmed overexpression of miR-223 in primary HSCs by qRT-PCR (Figure 1b). Overexpression levels of miR-223 were about 600-fold increase which should not be beyond the physiological range as a previous study has demonstrated >5000-fold reduction of miR-223 levels in human activated HSCs compared to their respective freshly isolated quiescent HSCs [14]. We observed no difference in morphology of control and miR-223 HSCs (Appendix A). Yet, overexpression of miR-223 caused downregulation of fibrotic gene transcripts including *COL1A1*, *COL3A1*, *LOXL2* and *ACTA2* (Figure 1c). A reduction in type I collagen production was further confirmed at the protein levels by immunostaining and Western blots (Figure 1d,e). Furthermore, we detected reduced type I collagen secretion from miR-223 HSCs (Figure 1f). Active proliferation is another hallmark of activated HSCs, and we found that overexpression of miR-223 reduced proliferation of HSCs as examined by the number of ki67^+^ cells and the MTT assay (Figure 1g,h). Consistently, the cell cycle genes such as *CCNE1* and *CCND1* were downregulated upon induction of miR-223 (Figure 1i). We overexpressed miR-223 in an independent line of primary human HSCs and obtained similar results (Appendix A). These data suggest that miR-223 suppressed activation features of human HSCs.

### 2.2. miR-223 Promotes Quiescent Features of Human HSCs

We next surveyed RNA expression of a panel of human HSC quiescent signature genes by qRT-PCR [36]. We found upregulation of several quiescent markers such as *GFAP*, *RELN*, *RSPO3*, *NGFR* and *HGF* upon miR-223 overexpression (Figure 2a). We confirmed upregulation of GFAP and RELN in miR-223 HSCs at the protein levels by immunofluorescence (Figure 2b). Quiescent HSCs is characterized by abundance of cytoplasmic lipid droplets [37], and so we explored ability to store lipids between control and miR-223 HSCs. Upon addition of palmitic acid (PA) in culture medium, we found that miR-223 promoted the presence of perinuclear lipid granules and enhanced levels of BODIPY staining (Figure 2c). Consistently, we observed higher induction of *PLIN2* and *PLIN3*, which encode lipid droplet proteins, in miR-223 HSCs than those of the control cells [38] (Figure 2d). Finally, we investigated ability of cells to store vitamin A (retinol), the hallmark of quiescent HSCs, by quantifying UV autofluorescence with flow cytometry, an established protocol to determine vitamin A-storing lipid droplets in HSCs [39,40]. We showed a greater shift in UV absorbance of miR-223 HSCs than that of control HSCs after supplementation of retinol into the culture medium (Figure 2e). Supporting this observation, we noticed higher expression levels of lecithin:retinol acyltransferase (LRAT), which catalyzes formation of retinyl esters in lipid droplets, in miR-223 HSCs compared with the control cells (Figure 2f). Taken together, we provided evidence that miR-223 can transcriptionally and functionally promote quiescent phenotypes in human HSCs.

### 2.3. miR-223 Suppressed Cytoskeletal Activity of Primary Human HSCs

HSCs are known to transform into myofibroblasts and increase their motility and contractility upon activation [4,5]. This transformation requires cytoskeletal reorganization which are responsible for regulation of HSC functions including proliferation and migration [41,42,43,44]. We explored if miR-223 plays a role in cytoskeletal dynamics of human HSCs. Using the scratch assay, we found that overexpression of miR-223 significantly impaired the migration ability to close the wound area (Figure 3a). Moreover, miR-223 reduced invasion of HSCs in response to chemoattractants as assessed by the Transwell assay (Figure 3b). To determine cell contractility, we embedded HSCs into three-dimensional collagen matrix and observed reduced contraction upon overexpression of miR-223 (Figure 3c). Similar defects in cytoskeletal activity mediated by miR-223 overexpression were observed in additional line of primary human HSCs (Appendix A). Together, our results indicate that miR-223 suppressed migration, invasion, and contractility of human HSCs.

### 2.4. miR-223 Directly Targets ACTA2 and Inhibits Erk1/2 and Smad2/3 Phosphorylation

Our data indicate that miR-223 may inhibit the reorganization of the cytoskeleton, where *ACTA2*, which encodes smooth muscle α-actin (α-SMA), is upregulated and plays a crucial role in myofibroblast migration and contraction [41,42,43] (Figure 3). These results prompted us to hypothesize that miR-223 may post-transcriptionally regulate expression of *ACTA2*. First, we confirmed the reduction of α-SMA through protein levels by immunofluorescence and Western blot (Figure 4a,b). Then, we performed bioinformatics analysis and found a putative binding site of miR-223-3p at the 3′ untranslated region (3′UTR) of *ACTA2* transcripts (Figure 4c). To demonstrate that miR-223-3p mediates translational repression, we employed a luciferase assay, where the reporter was fused to the 3′UTR of *ACTA2* and co-expressed with miR-223-3p. We observed a significant decrease in the luciferase signals in cells transfected with the 3′UTR fusion reporter compared with control cells transfected with the reporter lacking the sequence of 3′UTR (Figure 4d). Moreover, mutations of the *ACTA2* 3′UTR sequence hindered the binding of miR-223-3p, resulting in improved luciferase activity (Figure 4d). Previous animal studies of *ACTA2*-deficient HSCs reveal defects in mechanical signals leading to impaired Erk1/2 and Smad2/3 signal pathways [41,42]. Likewise, we observed reduced phosphorylation of Erk1/2 and Smad2 in primary human HSCs upon overexpression of miR-223 (Figure 4e,f). Consistently, nuclear translocation of Smad2 was impaired in miR-223 HSCs upon TGFβ stimulation (Figure 4g). Taken together, these results suggest that miR-223 directly inhibited *ACTA2* expression and impaired the cytoskeletal signaling in human HSCs.

### 2.5. Restoration of α-SMA Inhibits the Antifibrotic Effects of miR-223

To test if α-SMA depletion is partly responsible for the antifibrotic effects mediated by miR-223, we cloned the full-length open reading frame (ORF) of *ACTA2* sequence and expressed into miR-223 HSCs. Reconstitution of α-SMA levels was confirmed at both mRNA and protein levels (Figure 5a,b). We observed that ectopic expression of α-SMA can enhance fibrogenesis in miR-223 HSCs as indicated by increased levels of *COL1A1* and *COL3A1* transcripts as well as type I collagen by immunoblotting (Figure 5c,d). Proliferation rate of miR-223 HSCs was slightly improved but not statistically significant upon overexpression of α-SMA (Figure 5e). We also found that α-SMA can improve cytoskeletal activity of miR-223 HSCs including cell migration, invasion, and contraction (Figure 5f–h). In parallel with the retrieval of mechanical activity, Erk signal pathway was partially rescued in miR-223 HSCs with exogenous expression of α-SMA (Figure 5i). In addition, α-SMA significantly restored nuclear translocation of Smad2 in miR-223 HSCs upon TGFβ stimulation (Appendix A). Collectively, we demonstrated that miR-223 suppressed HSC activation partly via downregulation of *ACTA2*.

### 2.6. miR-223 Blocks Fibrogenesis Induced by Hepatotoxicity in Liver Organoid Models

Our data thus far have illustrated the protective roles of miR-223 against HSC activation. To address potential clinical relevance of this finding, we established hepatocyte-stellate cell-organoid cultures as a model for functional studies of miR-223 in human fibrosis induced by hepatotoxicity. The organoid culture system more faithfully recapitulates the pathophysiology of fibrosis than the conventional mono-layer culture [45]. We adapted a protocol for generation of hepatic organoids from previously published studies [36,45,46]. These organoids consist of primary hepatic stellate cells, either with or without miR-223 overexpression, and a human hepatoma cell line, Huh7 (Figure 6a). After self-assembly into the 3D structure, organoids containing control or miR-223 HSCs (dubbed as “control” and “miR-223” organoids, respectively) are morphologically indistinguishable and reveal distribution of HSCs toward the core of the organoids as previously described [36,46] (Figure 6b).

Acetaminophen (APAP) is a commonly used medication that can be converted into a toxic metabolite to induce hepatotoxicity and subsequently the fibrogenic response [47]. Although Huh7 cells may poorly metabolize APAP, this drug can induce cellular damage in hepatoma cells via distinct mechanisms independent of the cytochrome activity [48,49,50]. Accordingly, we found that APAP treatment for 24 h reduced organoid viability (Figure 6c), although overall morphology of the organoids remains similar (Appendix A). Under basal conditions, we observed less collagen deposition in miR-223 organoids than that in control organoids (Figure 6d), consistent with reduced expression and secretion of type I collagen upon miR-223 overexpression (Figure 1c–f). As expected, APAP significantly induced collagen production and fibrotic markers in organoids with control HSCs (Figure 6d,e). Yet, miR-223 organoids displayed a limited fibrogenic response at both mRNA and protein levels upon APAP exposure (Figure 6d,f). APAP did not affect viability nor expression of fibrotic markers in primary HSCs as previously reported, suggesting that the HSC activation is dependent on hepatocyte injury [46] (Appendix A).

In addition, we aimed to study fibrotic response in the co-cultured organoids by free fatty acid (FFA) exposure, which triggers steatosis and lipotoxicity [30]. This phenomenon is believed to play a significant role in tissue damage, fibrosis, and the development of non-alcoholic fatty liver disease (NAFLD) [51]. To promote steatosis in our organoid model, we supplemented palmitic acid (PA) into the culture medium and observed lipid droplet accumulation in organoids as well as increased activity of caspase 3 as an indication of hepatotoxicity [52] (Figure 6g,h). PA exposure induced deposition of collagen and activation of fibrotic markers in control organoids (Figure 6i and Appendix A). However, PA-induced fibrogenesis of miR-223 organoids was suppressed along with limited induction of fibrotic genes (Figure 6i and Appendix A). Overall, these results from 3D co-culture assays are strongly suggestive of a protective effect of miR-223 on hepatotoxicity-induced HSC activation and highlight the potential of miR-223 as a therapeutic target to control fibrosis.

## 3. Discussion

In this study, we demonstrated miR-223 as an antifibrotic factor that suppressed fibrotic markers and collagen production in primary human HSCs and co-cultured organoids. Reciprocally, miR-223 activated expression of quiescent HSC markers and promoted lipid re-accumulation and retinol storage in HSCs. Our mechanistic studies revealed that miR-223 negatively regulated cytoskeletal activity and fibrogenic cascades of HSCs via its direct binding with 3′ untranslated region (UTR) of *ACTA2* which encodes smooth muscle α-actin (α-SMA). Ectopic expression of α-SMA in miR-223-overexpressing HSCs could partially restore activation phenotypes. Finally, we illustrated the therapeutic potentials of miR-223 to control fibrosis in two human organoid models of liver injury. Together, we have integrated our data into a model describing how miR-223 suppresses HSC activation via modulating the actin cytoskeleton (Figure 7).

Previous studies have proposed that neutrophils and natural killer (NK) cells can suppress fibrogenesis via exosomal transfer of miR-223 to HSCs and proinflammatory macrophages [15,16,22,23]. Conversely, miR-223-knockout mice display greater fibrosis upon injury [15,16]. However, the direct roles of miR-223 in HSCs are still unclear, and all these prior studies are limited to rodents or transformed cells, which may not fully capture physiological relevance for humans. In this study, we employed primary human HSCs, the gold standard to study human liver fibrosis in vitro, in couple with organoid technology, to illustrate the antifibrotic roles of miR-223. Specifically, we showed that exogenous miR-223 is sufficient to suppress the activation of HSCs upon hepatoxicity within human hepatic organoids. These results are in line with previous animal studies and illustrate the utility of organoids as a preclinical model to validate functionality of therapeutic targets and study human disease mechanisms. Furthermore, our data warrant the development of in vivo methods to restore miR-223 expression in activated HSCs as a new antifibrotic strategy. Intriguingly, a recent study has described the therapeutic benefit of the synthetic miR-223 analog delivered to Kupffer cells in mitigation of liver inflammation and fibrosis [22]. Similar approach that enables delivery of miR-223 to HSCs could be synergistic to control fibrosis in various chronic liver diseases including NAFLD which currently lacks an effective antifibrotic treatment [53].

In addition to excessive collagen production, HSC activation is characterized by the development of a robust actin cytoskeleton. *ACTA2* encoding α-SMA is the prominent marker of activated HSC and plays a significant role in controlling cell motility and contractile force which can be converted into a chemical signal to promote expression of fibrotic genes [41,42,43,54,55,56]. Inhibition of α-SMA impairs ECM synthesis, cell contractility, and activity of Erk1/2, Smad2/3, and YAP/TAZ in myofibroblasts [41,42,43]. To date, post-transcriptional regulation of mechanical signals in HSCs has not been reported. Here, we identified direct interaction of miR-223 with 3′UTR of *ACTA2* transcripts as a novel mechanism to control mechanotransduction. To our knowledge, our study is the first to illustrate the regulatory roles of miRNAs that link cytoskeletal dynamics with the fibrogenic cascades. Testing the antifibrotic effect of miR-223 in *ACTA2*-deficient HSCs could further strengthen the interplay between miR-223 and α-SMA in regulating HSC activation. However, generation of knockout lines is challenging due to limited expansion potential of primary cells.

Apart from controlling cytoskeletal activity, miR-223 may play pleiotropic roles in biology of HSCs. Recent studies suggest that miR-223 may inhibit HSC activation via suppressing autophagy and hedgehog/PDGF signaling pathways [16,23]. Conversely, circular RNA PWWP2A was reported to enhance HSC activation and proliferation via sponging miR-223, subsequently increasing expression of Toll-like receptor 4 (TLR4), which induces the production of inflammatory cytokines [57]. While these studies focus on the role of miR-223 in suppressing HSC activation, we demonstrated for the first time that miR-223 could also transcriptionally and functionally promote the quiescent state of HSCs. In line with these results, miR-223 has been shown to the most enriched miRNA in quiescent human HSCs, suggesting its protective role against HSC activation or its involvement in HSC quiescence [14]. However, our data suggest that miR-223 does not fully induce all quiescent HSC features such as cell morphology. Although we showed that miR-223 downregulates expression of α-SMA, a key cytoskeletal protein in stellate cells, a previous study has shown that primary mouse HSCs with α-SMA deficiency display a similar size and shape as wild-type HSCs, suggesting that loss of α-SMA alone is not sufficient for morphological changes [42]. Mechanisms of how miR-223 induces or maintains inactive phenotypes of HSCs should be further investigated in future.

In summary, we have demonstrated a therapeutic potential for miR-223 in suppressing fibrosis in both primary human HSC and co-cultured organoid models. Not only did we show that miR-223 can reduce expression of fibrotic markers, proliferation, mobility, and contractility, but we also found that its overexpression promotes quiescent phenotypes of HSCs. The mechanism by which miR-223 functions is at least through reducing α-SMA expression which impairs the cytoskeletal signaling and subsequently fibrogenic pathways. Collectively, miR-223 represents an attractive target for antifibrotic therapy, and restoration of miR-223 in activated HSCs could be a viable approach to control liver fibrosis.

## 4. Materials and Methods

### 4.1. DNA Constructs and Lentiviral Production

The pre-miRNA sequence of miR-223 and full-length open reading frame (ORF) of *ACTA2* were amplified from human genomic DNA and cDNA of human hepatic stellate cells, respectively, and subsequently cloned into EcoRI/BamHI-cut pLV-EF1a-IRES-Puro (Addgene plasmid #85132). Correct insertion of DNA fragments was confirmed by Sanger sequencing. Lentivirus was generated in HEK293FT cells co-transfected with the second-generation packaging plasmids (pMD2.G, Addgene plasmid #12259 and psPAX2, Addgene plasmid #12260) and a transfer vector containing the gene of interest. Viral supernatants collected at 24 h and 48 h post transfection were pooled and filtered using a filter with a pore size of 0.45 μm prior to transduction into primary cells. All primers used for cloning can be found in Appendix A.

### 4.2. Cells and Cell Culture

Primary human hepatic stellate cells (pHSCs; lot #27742 and lot #25925) were purchased from ScienCell^TM^ Research Laboratories (San Diego, CA, USA) and propagated in culture medium (SteCM; catalog #5301) according to the manufacturer. Huh7 hepatoma cell line (JCRB0403) was obtained from the Japanese Collection of Research Bioresources (JCRB) Cell Bank (Osaka, Japan) and cultured in Dulbecco’s modified Eagle’s medium (DMEM) containing 10% fetal bovine serum (FBS) and 1% penicillin/streptomycin at 37 °C with 5% CO_2_. To overexpress gene of interest, pHSCs were transduced with lentivirus via spinoculation method (800 g for 40 min at 37 °C) and selected by puromycin (0.5 μg/mL) for 7 days before further analysis. For TGFβ stimulation, the cells were challenged with 10 ng/mL of TGFβ (Peprotech, Rehovot, Israel) for 6–24 h.

### 4.3. Organoid Culture

pHSCs and Huh7 cells were detached from two-dimensional (2D) cultures and mixed together in a ratio of 2:1 following the protocol adapted from previous studies [36,46]. From this mixture, ~10,000 cells were seeded into a 96-well round bottom ultra-low attachment microplate (Corning, Corning, NY, USA; catalog #7007). Within 48 h, the co-cultured organoids will be formed and maintained in a medium containing a ratio of 1:1 of Huh7 and pHSC culture medium. The co-cultured organoids were exposed to 15 mM acetaminophen (APAP) for 24 h or 300 μM palmitic acid (PA) for 48 h to induce hepatoxicity. These organoids were then harvested for subsequent analysis.

### 4.4. Immunofluorescence (IF) Staining and Microscopy

For the 2D culture, cells were fixed in 4% paraformaldehyde (PFA) for 15 min, blocked in 5% BSA with 0.05% Triton-X, and incubated overnight with primary antibody against type I collagen, α-SMA, GFAP and ki67 (1:1000; AbCam, Cambridge, MA, USA; catalog #ab34710, #ab5694, #ab7260, and #ab15580, respectively), Gata4 and Reelin (1:100; Santa Cruz, Dallas, TX, United States; catalog #sc-1237 and #sc-25346, respectively), followed by incubation with Alexa Fluor 488-conjugated or Alexa Fluor 594-conjugated secondary antibodies (Invitrogen™, Waltham, MA, USA; catalog #A-21206 and #A-11056). For the 3D culture, organoids were fixed in 4% PFA for 2 h, dehydrated in 30% sucrose overnight, embedded into Tissue Tek^®^ O.C.T. Compound, and sliced at 10 μm thickness in cryostat microtome. Slides with sectioned organoids were processed for antibody staining using the same method as cells from the 2D culture and preserved in Fluoromount-G™ Mounting Medium with DAPI (Invitrogen™; catalog #00-4959-52). Additional primary antibodies used for organoid studies include anti-albumin antibody (1:200; Bethyl, #A80-129A) and anti-PDGFRβ (1:1000, AbCam, catalog #ab69506). All IF samples were visualized by EVOS M7000 cell imaging system (Invitrogen™), and images at each fluorescent channel were taken at equivalent exposure for comparison.

### 4.5. Analysis of mRNA Expression

RNA isolation was performed by GenUP™ Total RNA Kit (Biotechrabbit, Berlin, Germany; Cat #BR0700903) or microRNA Purification Kit (Norgen Biotek, ON, Canada; Cat #21300). Reverse transcription was achieved by RevertAid First Strand cDNA Synthesis Kit (Thermo Scientific™; Cat #K1622, Carlsbad, CA, USA). Quantitative real-time PCR (qRT-PCR) was performed using the Applied Biosystems QuantStudio 5 Dx Real-Time PCR System (Thermo Scientific™) with CAPITAL^TM^ qPCR Green Mix HRox (biotechrabbit; catalog #BR0501902). Data were analyzed using the 2^−ΔΔCt^ method and normalized to *RPL19* expression. All primers used for qRT-PCR can be found in Appendix A.

### 4.6. Western Blot Analysis

Protein lysates were prepared in RIPA lysis buffer, separated by 12% SDS-PAGE and transferred to a nitrocellulose membrane (Cytiva, Marlborough, MA, USA; Cat #10600012). Then, membranes were incubated overnight with the following primary antibodies against type I collagen (1:500, AbCam, #ab34710), α-SMA (1:500, Abcam, #ab5694), Erk1/2 (1:1000, CST, #9102S), p-Erk1/2 (1:1000, CST, #4370S), Smad2 (1:1000, CST, #5339S), p-Smad2, (1:1000, CST, #3108S) and GAPDH (1:2000, Invitrogen, #437000). HRP-conjugated goat anti-mouse IgG and goat anti-rabbit IgG (1:5000, CST, #7076S and #7074, respectively) were used as secondary antibodies. Blots were scanned using a UVP ChemStudio instrument (Analytik Jena, Beverly, MA, USA). Uncropped blots are provided in Appendix A.

### 4.7. Analysis of Cell Proliferation and Apoptosis

The proliferation and viability of cells were determined using an MTT assay (Invitrogen™, catalog #M6494), and detection of apoptosis was accomplished by EnzChek™ Caspase-3 Activity Assay Kits (Invitrogen™, catalog # E13183) according to the manufacturer’s protocols.

### 4.8. Cell Motility and Contraction Assays

For a migration assay, a cell monolayer of HSCs was scratched with a p10 pipette tip. Images were captured with a light microscope at different time points, and would closure distances were measured by ImageJ software. For an invasion assay, the Transwell^®^ insert with pore size of 8.0 μm (Corning; cat #CLS3422) was coated with Matrigel^®^ matrix (Corning; cat #356234) and seeded with HSCs in the serum-free SteCM medium. Complete SteCM medium was added to the basolateral chamber as a chemotactic agent. After 24–48 h, cells that remain in the apical chamber were removed with cotton swabs, while the cells that migrated through the membrane were stained with DAPI and visualized by EVOS M7000 cell imaging system. For a contraction assay, ~1–2 × 10^5^ HSCs were trypsinized and embedded into 1 mg/mL collagen matrices (Sigma-Aldrich, St. Louis, MO, USA; Cat #08-115) in a well of a 24-well plate. After solidifying, the gel was dislodged from the well by gently running the tip of a 200 μL pipet tip along the gel edge. Contraction of the gel was recorded by a digital camera to obtain images at 12 h intervals from 0 to 36 h.

### 4.9. Hydroxyproline Assay

Hydroxyproline levels in culture supernatants were determined using a hydroxyproline assay kit (Sigma-Aldrich; Cat #MAK008) according to the manufacturer’s instructions. Hydroxyproline content was expressed as μg/mL of culture supernatants.

### 4.10. Prediction of MiRNA Targets and Dual-Luciferase^®^ Reporter Assay

The hybridization pattern between miR-223 and 3′UTR of *ACTA2* was predicted by using RNA hybrid tool [58]. Wild-type and mutant 3′UTR sequences of *ACTA2* were cloned into SacI/XhoI-cut pmirGLO vector (Promega, Madision, WI, USA; Cat #E133A). The sequence of miR-223-3p was cloned into BamHI/HindIII-cut pSilencer 3.0 H1 (Thermo Scientific; Cat #AM7210). Primers used for the cloning can be found in Appendix A. The interaction between miR-223 and its target was tested by the Dual-Luciferase^®^ Reporter Assay System (Promega; Cat #E1910) by following the manufacturer’s instruction. Firefly luciferase activity is normalized against Renilla luciferase.

### 4.11. Lipid Droplet and Vitamin A Storage Assay

Primary HSCs were incubated with SteCM medium supplemented with palmitic acid (100 μM; Sigma-Aldrich; Cat #P9767) for three days to induce lipid droplet formation followed by BODIPY staining (Sigma-Aldrich; Cat #790389). Vitamin A content of primary HSC was indirectly determined by measuring the auto-fluorescence after UV light excitation (retinyl ester auto-fluorescence at 328 nm) using a MACSQuant^®^ X Flow Cytometer (Miltenyi Biotec, Gaithersburg, MD, USA). The shift in UV intensity was evaluated in pHSC after incubation with SteCM medium with or without retinol (5 μM) for 24 h.

### 4.12. Statistical Analysis

All data were presented as the mean ± standard deviation of the mean (SD) and analyzed using GraphPad Prism software version 6.01 (GraphPad, San Diego, CA, USA). Significant differences between mean values were performed at least in three triplicates from at least two independent experiments. Differences between two groups were compared using two-tailed unpaired Student’s *t*-test analysis. A *p* value of  <0.05 was considered statistically significant.

## Figures and Tables

**Figure 1 ijms-23-09380-f001:**
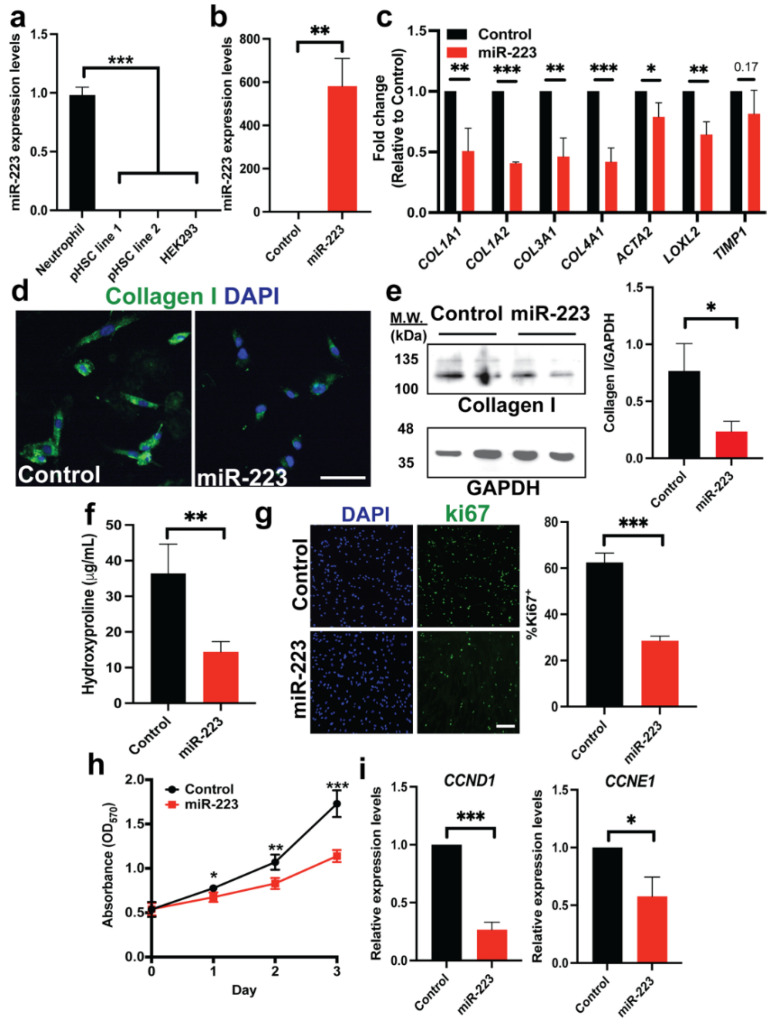
miR-223 suppresses expression of fibrotic markers and proliferation in human HSCs. (**a**) Real-time quantitative PCR (RT-qPCR) of miR-223 expression in human neutrophils (HL-60; positive control), two independent lines of primary human HSCs (pHSCs), and HEK293T (negative control). (**b**) Lentivirus-mediated overexpression of miR-223 in pHSCs measured by qRT-PCR. (**c**) Expression of fibrotic gene transcripts upon overexpression of miR-223 determined by qRT-PCR. (**d**,**e**) Expression of type I collagen in control and miR-223 pHSCs by immunofluorescence and Western blot. (**f**) Pro-collagen secretion of control and miR-223 pHSCs into culture medium determined by hydroxyproline assay. (**g**,**h**) Rate of HSC proliferation as examined by the number of ki67^+^ cells and the MTT assay. (**i**) Expression levels of cell cycle genes *CCND1* and *CCNE1* by qRT-PCR. Data presented as the mean ± SD and expressed relative to those of control HSCs (set as 1.0) for qRT-PCR experiments. N = 3–4 replicates per group from at least two independent experiments. Student’s *t*-test; * = *p* < 0.05, ** = *p* < 0.01 and *** = *p* < 0.001. Fluorescent images taken at equivalent exposure for comparison. Scale bars, 100 μm. DAPI: 4′,6-diamidino-2-phenylindole. MTT: 3-(4,5-dimethylthiazol-2-yl)-2,5-diphenyl-2H-tetrazolium bromide.

**Figure 2 ijms-23-09380-f002:**
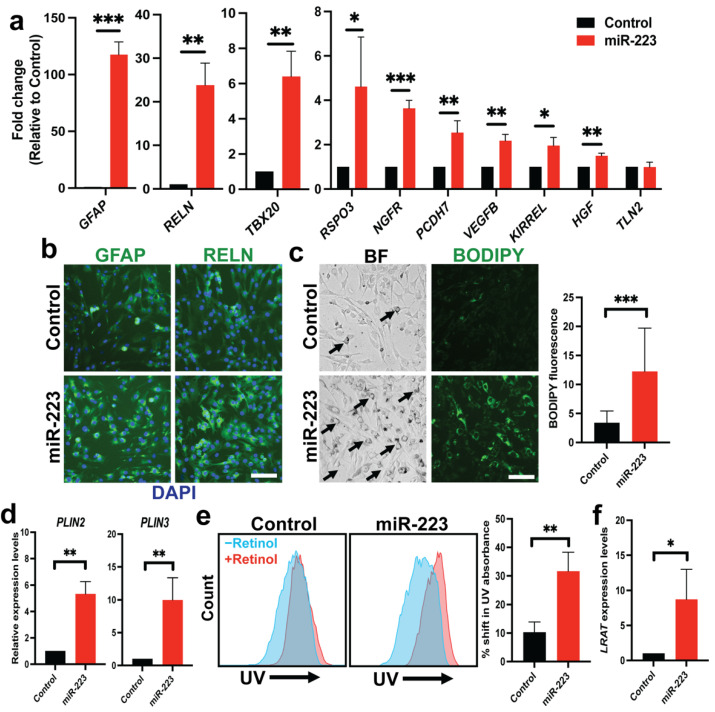
miR-223 promotes quiescent phenotypes of human HSCs. (**a**) RNA expression of quiescent HSC genes upon overexpression of miR-223 determined by qRT-PCR. (**b**) Expression of GFAP and RELN in control and miR-223 HSCs by immunofluorescence. (**c**) Formation of cytoplasmic lipid droplets upon incubation with palmitic acid (100 μM) in control and miR-223 HSCs visualized by bright field (BF) microscopy and BODIPY staining. Black arrows indicate cells with perinuclear lipid droplets. (**d**) Expression levels of genes that encode lipid droplet proteins *PLIN2* and *PLIN3* by qRT-PCR. (**e**) UV autofluorescence of control and miR-223 HSCs determined by flow cytometry. (**f**) RNA expression levels of lecithin:retinol acyltransferase (LRAT) by qRT-PCR. Data presented as the mean ± SD and expressed relative to those of control HSCs (set as 1.0) for qRT-PCR experiments. N = 3–4 replicates per group from at least two independent experiments. Student’s *t*-test; * = *p* < 0.05, ** = *p* < 0.01 and *** = *p* < 0.001. Fluorescent images taken at equivalent exposure for comparison. Scale bars, 100 μm. DAPI: 4′,6-diamidino-2-phenylindole. BODIPY: 4,4-Difluoro-1,3,5,7,8-Pentamethyl-4-Bora-3a,4a-Diaza-s-Indacene. UV: ultraviolet.

**Figure 3 ijms-23-09380-f003:**
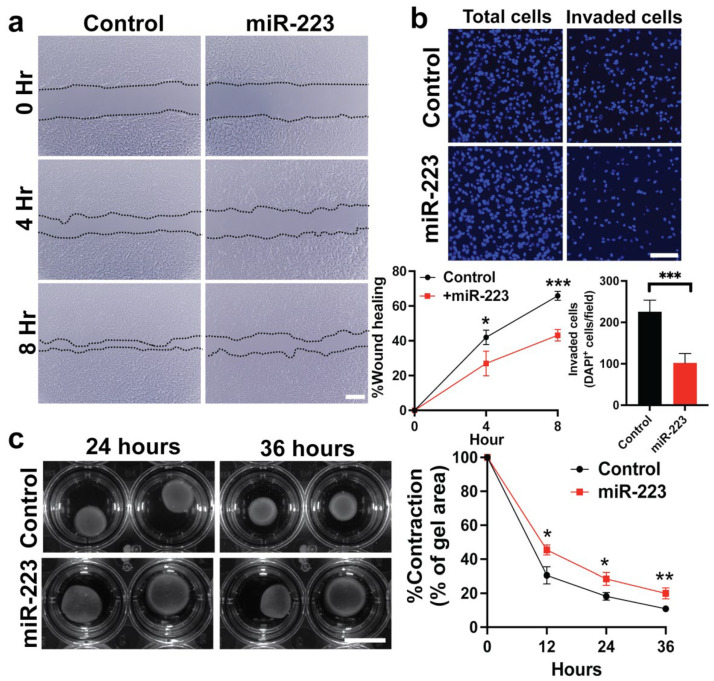
miR-223 suppresses migration, invasion, and contractility of primary human HSCs. (**a**) Representative images of scratch assay to determine ability for wound closure of control and miR-223 HSCs at different time points. Dash lines indicate the border of migration. (**b**) Representative fluorescent images of nuclear DAPI staining indicated invaded cells after seeding in a Matrigel-coated Transwell for 24 h with chemoattractant (2% FBS). Invaded cells were counted from 10 random microscope fields for each group. (**c**) Representative images of collagen matrix contraction at 24 and 36 h after the lattices were dislodged. Gel contraction was measured at 12 h intervals as indicated. Data presented as the mean ± SD. N = 3–4 replicates from at least two independent experiments. Student’s *t*-test; * = *p* < 0.05, ** = *p* < 0.01 and *** = *p* < 0.001. Scale bars, 200 μm (**a**), 100 μm (**b**), and 5 mm (**c**). DAPI: 4′,6-diamidino-2-phenylindole.

**Figure 4 ijms-23-09380-f004:**
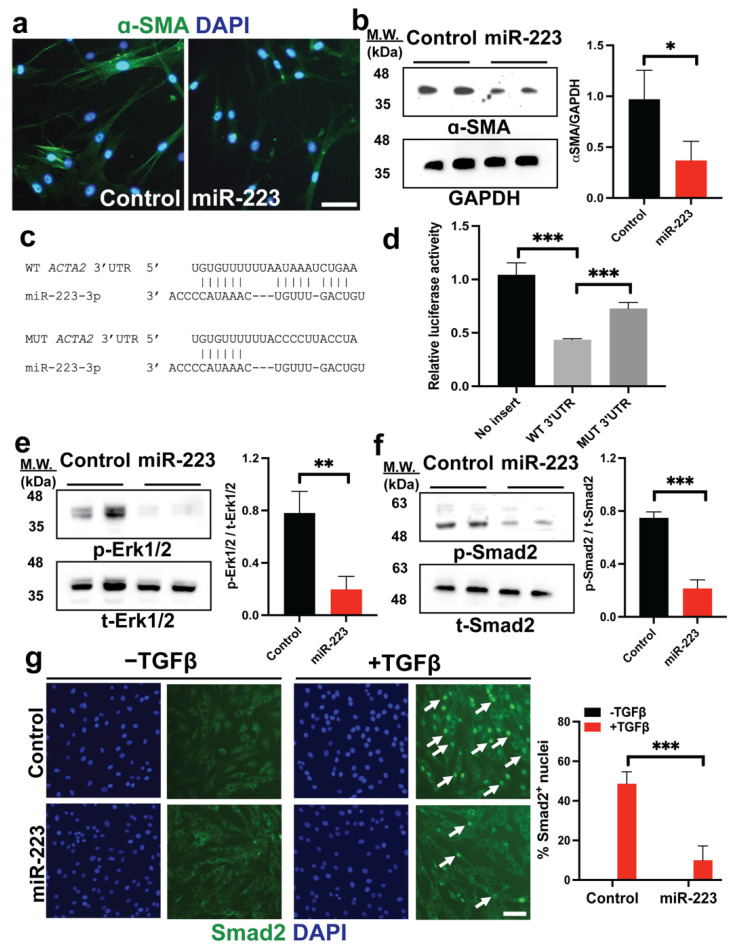
miR-223 directly targets *ACTA2* and impairs fibrogenic signal pathways (**a**,**b**) Expression of alpha-smooth muscle actin (α-SMA) in control and miR-223 pHSCs by immunofluorescence and Western blot. (**c**) RNA hybridization between miR-223-3p and the wild-type (WT) or mutant (MUT) sequence of 3′ untranslated region (3′UTR) of *ACTA2* transcripts. (**d**) Dual luciferase reporter assay determining direct interaction between miR-223 and *ACTA2* 3′UTRs by levels of luciferase activity. (**e**) Levels of phosphorylated Erk1/2 in control and miR-223 HSCs determined by Western blot in basal medium conditions. (**f**) Levels of phosphorylated Smad2 in control and miR-223 HSCs determined by Western blot after TGFβ exposure. (**g**) Immunofluorescence of Smad2 for control and miR-223 HSCs before and after TGFβ stimulation. White arrows indicate cells with nuclear translocation of Smad2. Quantification of nuclear Smad2^+^ cells were counted from 10 random microscope fields for each group. Data presented as the mean ± SD. N = 3–4 replicates from at least two independent experiments. Student’s *t*-test; * = *p* < 0.05, ** = *p* < 0.01 and *** = *p* < 0.001. Scale bars, 100 μm. DAPI: 4′,6-diamidino-2-phenylindole.

**Figure 5 ijms-23-09380-f005:**
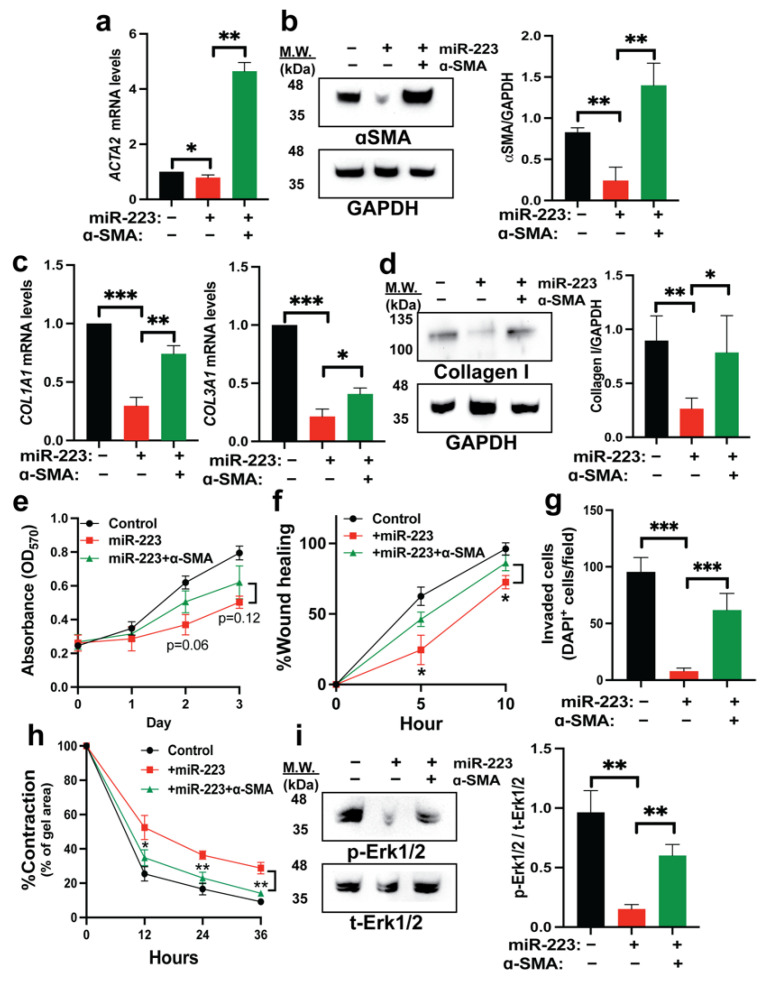
Reconstitution of α-SMA partially restores activation phenotypes of miR-223-overexpressing HSCs. (**a**,**b**) Expression of alpha-smooth muscle actin (α-SMA) in control, miR-223, miR-223 with α-SMA overexpression HSCs by immunofluorescence and Western blot. (**c**) Expression levels of *COL1A1* and *COL3A1* mRNAs upon α-SMA reconstitution by qRT-PCR. (**d**) Protein expression of type I collagen upon α-SMA reconstitution by Western blot. (**e**) Rate of HSC proliferation as determined by the MTT assay. (**f**) Scratch assay to determine ability of cell migration. (**g**) Quantification of invaded cell number after seeding in a Matrigel-coated Transwell for 24 h with chemoattractant (2% FBS). Invaded cells were counted from 10 random microscope fields for each group. (**h**) Gel contraction assay measured at 12 h intervals after the lattices were dislodged. (**i**) Levels of phosphorylated ERK1/2 upon α-SMA reconstitution determined by Western blot in basal medium conditions. Data presented as the mean ± SD and expressed relative to those of control HSCs (set as 1.0) for qRT-PCR experiments. N = 3–4 replicates from at least two independent experiments. Student’s *t*-test; * = *p* < 0.05, ** = *p* < 0.01 and *** = *p* < 0.001.

**Figure 6 ijms-23-09380-f006:**
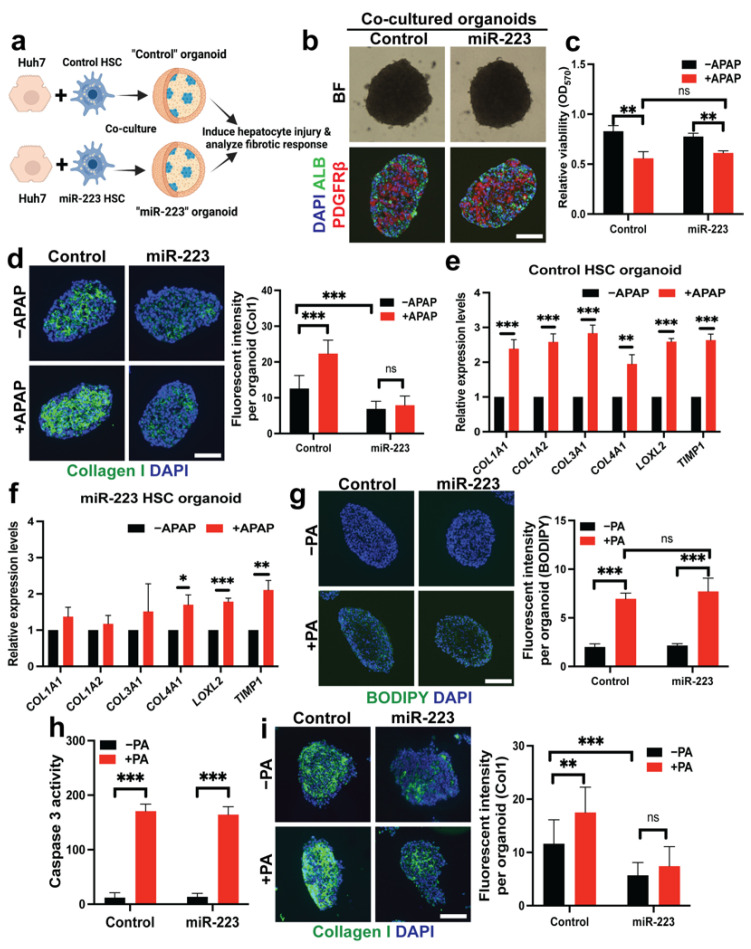
miR-223 suppresses collagen production in injury models of human hepatic organoids. (**a**) Schematic representation of experimental design to generate co-cultured organoids with hepatoma cells, Huh7, and primary HSCs (pHSCs) with or without miR-223 overexpression. (**b**) Representative brightfield (BF) and fluorescent images of co-cultured organoids containing hepatoma cells (ALB^+^) and pHSC (PDGFRβ^+^). (**c**) Viability of hepatic organoids with or without APAP treatment as determined by MTT assay. (**d**) Representative fluorescent images of hepatic organoids with type I collagen stain in the model of APAP-induced liver injury. (**e**,**f**) Levels of fibrotic gene transcripts from co-cultured organoids as determined by qRT-PCR. (**g**) BODIPY staining of hepatic organoids after incubation with 300 μM of palmitic acid (PA) for 48 h. (**h**) Apoptosis of hepatic organoids with or without PA treatment as determined by caspase-3 activity. (**i**) Representative fluorescent images of hepatic organoids with type I collagen stain in the model of PA-induced liver injury. Data presented as the mean ± SD expressed relative to those of untreated organoids (set as 1.0) for qRT-PCR experiments. Quantification of fluorescent levels were calculated from at least 10 random organoids for each group from three independent experiments (**d**,**g**,**i**). N = 3–4 replicates from at least two independent experiments (**e**,**f**,**h**). Fluorescent images taken at equivalent exposure. Student’s *t*-test; * = *p* < 0.05, ** = *p* < 0.01 and *** = *p* < 0.001. Scale bars, 100 μm. APAP: N-acetyl-para-aminophenol. DAPI: 4′,6-diamidino-2-phenylindole. ALB: albumin. BODIPY: 4,4-Difluoro-1,3,5,7,8-Pentamethyl-4-Bora-3a,4a-Diaza-s-Indacene.

**Figure 7 ijms-23-09380-f007:**
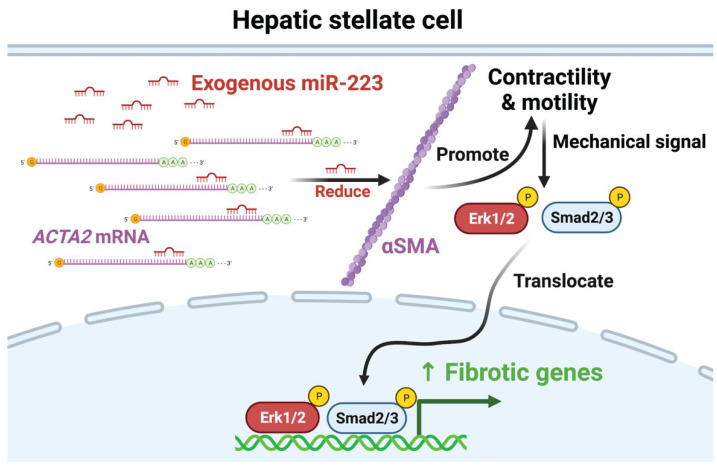
miR-223 suppresses HSC activation via modulating the actin cytoskeleton. Exogenous miR-223 can directly bind the 3′UTR of *ACTA2* transcripts, leading to reduced protein levels of α-SMA. Impaired cytoskeletal activity then attenuates fibrogenic signaling pathways, including phosphorylation of Erk1/2 and Smad2, and reduces expression of fibrotic genes.

## Data Availability

All data generated or analyzed during this study are included in this published article and its Appendix A. Raw data of all quantitatively analyzed experiments are available from the corresponding author on reasonable request.

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
