# Peer review of "MicroRNA-223 Suppresses Human Hepatic Stellate Cell Activation Partly via Regulating the Actin Cytoskeleton and Alleviates Fibrosis in Organoid Models of Liver Injury"

_ijms, 2022, doi:10.3390/ijms23169380_

Round 1
Reviewer 1 Report
In this study, the authors investigated the roles of miR-223 in the activation of primary human HSCs and tried to understand the mechanism of miR-223 action. There are several issues that need to be addressed.
Major concerns
1. To test the anti-fibrotic effect of miR-223, they expressed miR-223 in HSCs via lentiviral gene. Though the fibrotic gene transcripts including COL1A1, COL3A1, LOXL2 and ACTA2 were found downregulated (Fig. 1c) and quiescent markers upregulated (Fig. 2A), the authors explained that there was no difference in morphology between control and miR-223 HSCs. The authors need to provide a suitable reason for this, since the activation/inactivation of HSCs comes with drastic morphological changes.
2. The authors demonstrated that ACTA2 may be another target of miR-223 (Fig 4c, d). To test if a-SMA depletion is partly responsible for the miR-223-mediated antifibrotic effects, a-SMA was overexpressed in miR-223 HSCs (Fig. 5c-). However, as the overexpression of a-SMA itself undoubtedly brings dramatic morphological/biochemical changes (activation), the authors need to design another experiment which decisively proves the role of a-SMA in the miR-223 action.
Minor concern
The authors showed that the levels of phosphorylated Smad2 and its nuclear translocation were reduced in miR-223-overexpressing HSCs compared to their control. How does the miR-223 expression affect Smad3?
Reviewer 2 Report
In this manuscript, Aryachet et al. describe a therapeutic potential for miR-223 in suppressing fibrosis in both primary human HSC and co-cultured organoid models. They show that miR-223 can reduce expression of fibrotic markers, proliferation, mobility, and contractility, also that its over expression promotes quiescent phenotypes of HSCs.
The authors performed several experiments using primary HSCs (pHSCs) and miR-223 overexpressed HSCs (miR-223 HSCs). However, this reviewer has the following concerns.
Major comments:
1. The expression level of miR-223 in miR-223 HSCs seems 600-fold higher than that in pHSCs (Figure 1b). This level seems over physiological level and cytotoxic effects might appear.
2. The possible off-target effect of miR-223 should be discussed.
3. The expression level of ACTA2 (supposed major target gene of miR-223) is less suppressed than other fibrotic genes in mif-223 HSCs (Figure 1c).
4. Why are expression levels of the GAPDH gene different between control pHSCs (Figure 1e)?
5. The protein expression level of a-SMA in miR-223 HSCs is about half of that in pHSCs (Figure 4b, Figure 5b), although the mRNA expression level of the ACTA2 gene is similar between pHSCs and miR-223 HSCs (Figure 1c, Figure 5a). Why does this discrepancy occur?
Reviewer 3 Report
The present study is one more interesting, necessary addition to current understanding of the critical role of miR-223 in liver physiology and pathogenesis. The present study aimed to elucidate the pivotal role of miR-223 in negative regulation of the profibrotic phenotype of primary human HSCs. HSCs are the dominant effector of liver fibrosis through the complex interplays with resident parenchymal and nonparenchymal cells and progenitor cells of liver, immune cells, and bone marrow-derived progenitor cells.
In the text, facts known about miR-223 are addressed. Key models including the hepatic organoid model are concisely addressed. The present experiments have clearly dissected the interplays including the direct interaction of miR-223 with 3’UTR of ACTA2 transcripts as a novel mechanism to control mechanotransduction, in contrast to the control, as appropriate and feasible.
Suggestions:
-It is uncommon to place conclusive content at the end of Introduction section. Please move and incorporate lines 87–95 elsewhere appropriate.
-Researchers worldwide have been working on the complex interplays between HSCs and others including LSECs in terms of liver fibrosis progression and regression. Please concisely address and converge relevant issues into no more than 3 sentences in Introduction section.
Round 2
Reviewer 1 Report
Therapeutic effects of miR-223 was tested by the authors using liver organoid model and also by other groups using animal models of liver fibrosis. In this study, the authors tried to elucidate the mode of anti-fibrotic action of miR-223 which reportedly has many (>20) target genes.
1. The authors showed that miR-223 overexpression led to a reduction in alpha-SMA expression in commercially available human hepatic stellate cell line and thereby suppressed migration, invasion, and contractibility, indicating that miR-223-induced alpha-SMA regulated cytoskeletal organization. They, however, explained that there was no difference in morphology between control and miR-223 HSCs. In response to this problem, they hypothesized that miR-223 HSCs are partially inactivated and thus may not fully acquire all features of quiescent HSCs including morphological changes. But this cannot be an adequate answer to the problem since throughout the manuscript they attributed the anti-fibrotic action of miR-223 to the loss of alpha-SMA. To address the problem, the authors need to perform the experiments using the tissue-derived stellate cells instead of the cell line used.
2. Even if miR-223 exerts its anti-fibrotic effect via the suppression of other target(s) than alpha-SMA, the overexpression of alpha-SMA itself likely incurs dramatic changes (toward the activation) in miR-223 HSCs, and this experiment may not determine which gene(s) is the main culprit. With that in mind, the experimental data in Fig 5 is not sufficient to prove the role of alpha-SMA. The authors need to provide critical data for the role of alpha-SMA in the miR-223 action, which is the main theme of the manuscript.
Reviewer 2 Report
All comments are resolved.
Round 3
Reviewer 1 Report
unseen